# Structural basis for the activation of a compact CRISPR-Cas13 nuclease

Xiangyu Deng[1,5], Emmanuel Osikpa [2,5], Jie Yang [2], Seye J. Oladeji [1], Jamie Smith[1], Xue Gao [2,3,4] ✉ & Yang Gao [1] ✉

The CRISPR-Cas13 ribonucleases have been widely applied for RNA knockdown and transcriptional modulation owing to their high programmability and specificity. However, the large size of Cas13 effectors and their non-specific RNA cleavage upon target activation limit the adeno-associated virus based delivery of Cas13 systems for therapeutic applications. Herein, we report detailed biochemical and structural characterizations of a compact Cas13 (Cas13bt3) suitable for adeno-associated virus delivery. Distinct from many other Cas13 systems, Cas13bt3 cleaves the target and other nonspecific RNA at internal "UC" sites and is activated in a target length-dependent manner. The cryo-electron microscope structure of Cas13bt3 in a fully active state illustrates the structural basis of Cas13bt3 activation. Guided by the structure, we obtain engineered Cas13bt3 variants with minimal off-target cleavage yet maintained target cleavage activities. In conclusion, our biochemical and structural data illustrate a distinct mechanism for Cas13bt3 activation and guide the engineering of Cas13bt3 applications.

The CRISPR-Cas13 system is an RNA-guided ribonuclease with a single Cas13 protein effector and a CRISPR RNA (crRNA)[1,2]. Upon proper target RNA binding and pairing with the spacer segment of the crRNA, the Higher Eukaryotes and Prokaryotes Nucleotide-binding (HEPN) nuclease domains within Cas13 are activated to cleave the bound RNA in cis (on-target cleavage activity) and any surrounding non-specific single-strand RNA (ssRNA) in trans (collateral activity)[2,3]. Cas13 has been widely adapted for in vivo RNA knockdown and transcriptome modulation[2,4–9], due to its high efficiency and programmability in RNA targeting. However, the in vivo applications of CRISPR-Cas13 systems are hindered by non-specific cleavage of the activated Cas13, which could damage the transcriptome and cause cell death[2,3,10–12]. In addition, the large size of Cas13 effectors (967–1152 amino acids) limits the possibility of packaging them into adeno-associated viruses and subsequent in vivo delivery for therapeutic applications[13,14]. A deep mechanistic understanding of the Cas13 nuclease activation and cleavage will be critical to modulate the Cas13 collateral activities and reduce the effector size for in vivo applications.

Four subtypes of Cas13 (a–d) have originally been characterized, each having its own unique domain organization and crRNA sequence, exhibiting different levels of on-target and collateral cleavage activities[1,4,15–17]. The structural basis of Cas13 HEPN nuclease activation has been investigated for several Cas13 systems[18–24]. In both Cas13a and Cas13d, target binding induces inter-domain conformational changes and brings the two catalytic HEPN domains closer to each other to allow cleavage of both the target and nonspecific RNAs[20,21]. Recently, a series of compact Cas13 proteins (~800 amino acids) suitable for in vivo applications have been reported[7,25,26] and structures of the binary and ternary complexes of the compact *Planctomycetes bacterium* Cas13bt3[25] (also known as Cas13X.1[7]) were determined[27]. Interestingly, the HEPN1 and HEPN2 domains of Cas13bt3 move in unison over 24 Å upon target binding. However, the two HEPN domains adopted the same conformation relative to each other in both structures[27], and how this rigid body movement of the HEPN domains induces nuclease activation was unclear. Moreover, the ternary complex of Cas13bt3 was prepared with a short, 25 nucleotide (nt) target RNA[27], which does not

[1]Department of BioSciences, Rice University, Houston, TX 77005, USA. [2]Department of Chemical and Biomolecular Engineering, Rice University, Houston, TX 77005, USA. [3]Department of Bioengineering, Rice University, Houston, TX 77005, USA. [4]Department of Chemistry, Rice University, Houston, TX 77005, USA. [5]These authors contributed equally: Xiangyu Deng, Emmanuel Osikpa. ✉e-mail: xue.gao@rice.edu; yg60@rice.edu

support optimal Cas13bt3 activation[7]. Further structural and mechanistic studies are essential to illustrate the molecular basis of Cas13bt3 activation and guide the engineering of Cas13bt3 for in vivo applications.

Here, we illustrate the allosteric activation mechanism of Cas13bt3, which guides the rational engineering of a highly specific Cas13bt3 variant. Biochemically, Cas13bt3 cleaves target and nonspecific RNA at internal "UC" sites and displays target length-dependent activation. A structure of the activated Cas13bt3 in complex with crRNA and 30 nt target RNA was determined at 3.5 Å with a cryo-electron microscope (cryo-EM). Compared to previous structures, the HEPN nuclease domains exhibit almost 50 Å of target length-dependent translocation along the target RNA. Coupled with the movement of HEPN domains, two inter-domain linkers (IDLs) that bridge the HEPN domains and crRNA binding domains become fully stretched and form new interfaces critical for target RNA binding and Cas13bt3 activation. Based on the proposed allosteric mechanism, mutagenesis on or around the IDLs produced new Cas13bt3 variants with minimal collateral activity while maintaining target cleavage activities in biochemical assays and mammalian cells. We expect the similar mechanism and engineering strategies to be applicable to other Cas13 systems.

## Results
### Biochemical characterization of Cas13bt3 ribonuclease activity
Cas13bt3 was reported to have crRNA-guided endonuclease activity against target and nonspecific ssRNA (Fig. 1a). According to the Cas13bt3-associated CRISPR array from the uncultured microorganism genome[7,25], we designed a crRNA with a 30 nt spacer for testing

Cas13bt3 activities against target and nonspecific RNA. Cas13bt3 is inactive in the absence of a crRNA whereas crRNA binding activates Cas13bt3 to efficiently cleave a 122 nt target RNA (T0) (Fig. 1b). Interestingly, the activated Cas13bt3 complex (Cas13bt3 complex with the crRNA and a fully matched target RNA (T1)) shows low collateral cleavage activity and a pattern of exonuclease cleavage against a poly-U reporter RNA (U20, Fig. 1c, f), which is the favored substrate for several other types of Cas13[3,4,15,28]. In contrast, the activated Cas13bt3 efficiently cut a 36 nt ssRNA (ssRNA36), likely around a "UUC" sequence present in the ssRNA36 (Supplementary Fig. 1a, b). We then designed a series of short 10 nt ssRNA reporters each with one nucleotide variation (R1-5, Fig. 1f). Cas13bt3 efficiently cleaves the R1 substrate at "UUC" sites with electrophoresis assay as well as in a fluorescent assay with the substrate harboring a 5′-fluorophore and a 3′-quencher (R1-FQ) (Fig. 1d, e, f, Supplementary Fig. 1c). To systemically probe the substrate preference of Cas13bt3, we further demonstrated that poly-A10, poly-C10, and poly-G10 are uncleavable by Cas13bt3 compared to R1 RNA, and U10 showed the same pattern of low exonuclease cleavage as the poly-U20 reporter RNA (Fig. 1c, Supplementary Fig. 1d). We then put the 16 possible combinations on AAANNAAAAA substrates (N refers to A, U, G, or C) and found that only the substrate with the UC motif can be efficiently cleaved by activated Cas13bt3 (Fig. 1g). In addition, we confirmed that nucleotides surrounding the "UC" motif do not affect Cas13bt3 activity (Supplementary Fig. 1e).

Next, we measured the target length dependence of Cas13bt3 activation and cleavage. We first varied the target length from 30 nt (T1) to 27 (T2), 24 (T3), and 21 (T4) nt, which exhibited 67%, 42%, and 6% collateral cleavage activity relative to the activity of T1 (Fig. 2a, b). We then kept the target length as 30 nt but gradually introduced

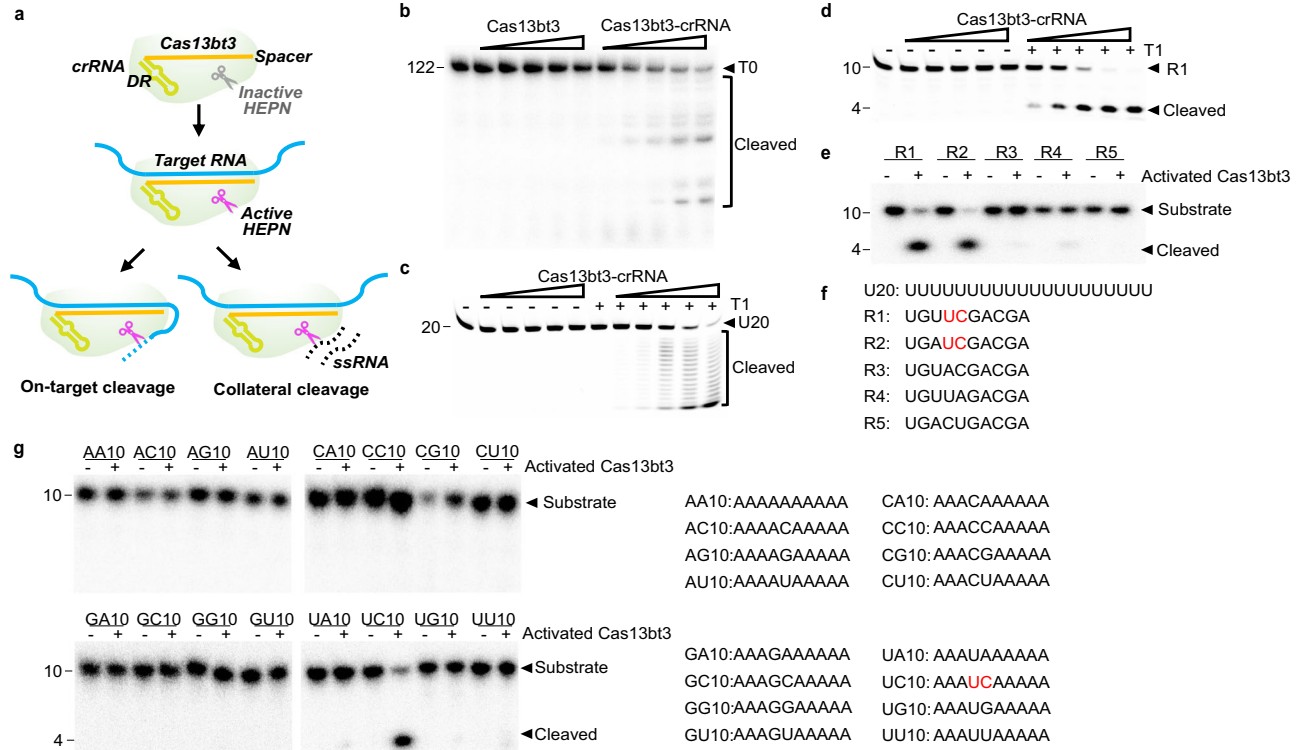

**Fig. 1 | Biochemical characterization of Cas13bt3 cleavage. a** Diagram of on-target and collateral activity of Cas13bt3. **b** On-target cleavage of a 122 nt target RNA (T0) by Cas13bt3 in the presence and absence of crRNA. The T0 substrate and the cleavage products were indicated. The Cas13bt3-crRNA concentrations are 12.5–200 nM. **c, d** Collateral cleavage of the U20 substrate (**c**) and the R1 substrate (**d**) by 12.5-200 nM of activated Cas13bt3 complex. **e** The activated Cas13bt3 collateral cleavage of reporter RNAs with varied sequences (R1-R5), the concentration of activated Cas13bt3 is 50 nM. **f** Sequences of the reporter RNAs used in the collateral cleavage assay. **g** The activated Cas13bt3 collateral cleavage of reporter RNAs with 16 possible combinations, the concentration of activated Cas13bt3 is 50 nM, and the sequences of the reporter RNAs are shown on the right panel. **f, g** The UC cleavage site is highlighted in red. (**b**–**e**, **g**) RNA sizes (in nt) are indicated. At least three times each experiment was repeated independently with similar results. Source data are provided as Source Data File.

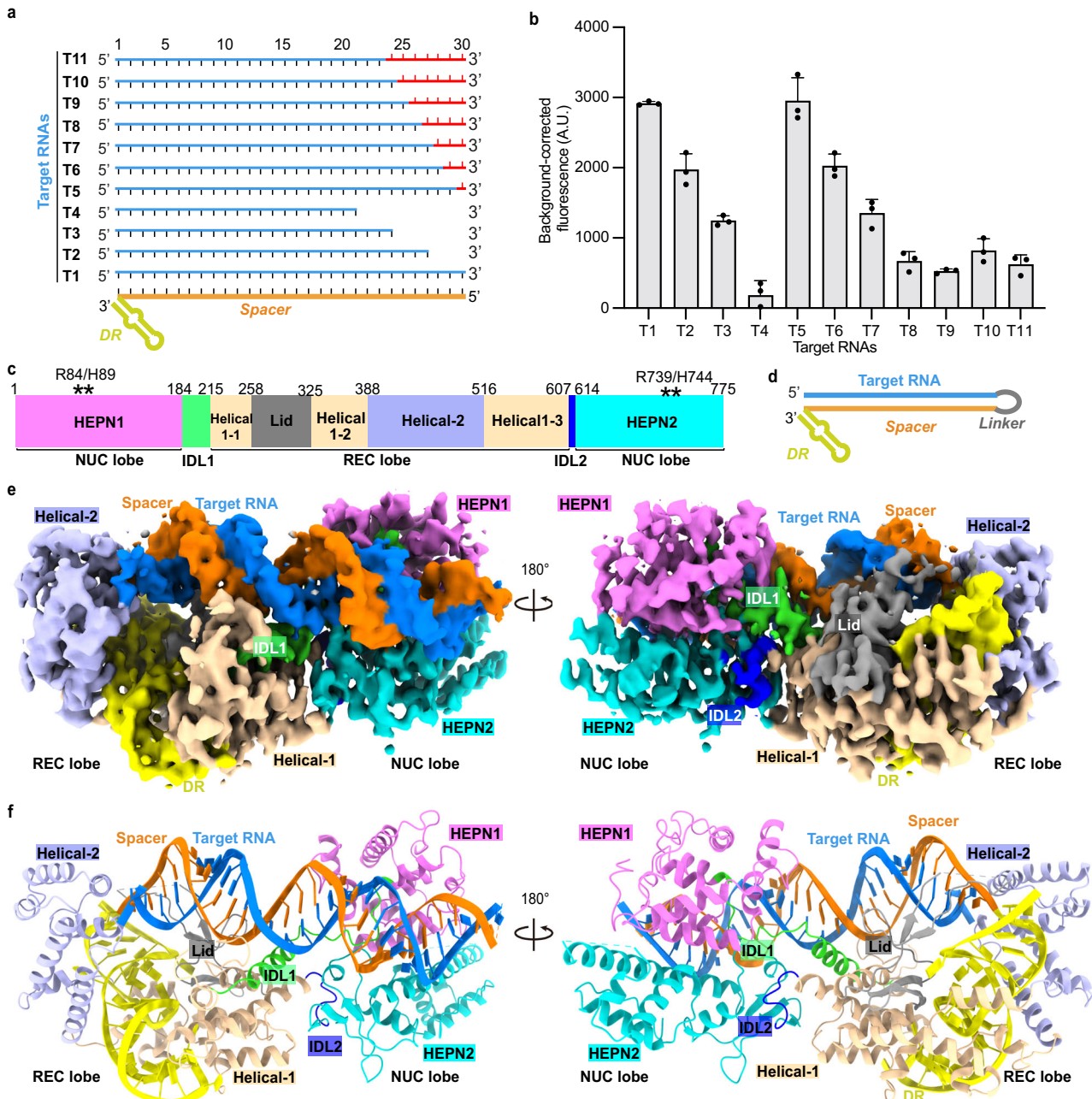

**Fig. 2 | Cryo-EM structure of activated Cas13bt3-hpRNA complex. a** Substrate design for examining target-length dependence of Cas13bt3 cleavage. The target length from 5′- to 3′-ends is indicated by the scale bar on the top. The mismatched RNA segment is colored in red. **b** Collateral cleavage activity of Cas13bt3 with different target RNAs in (**a**). R1-FQ was used as reporter RNA. Data are presented as means ± SDs (*n* = 3). A.U. relative fluorescence intensity in arbitrary units. **c** Domain structures of Cas13bt3. **d** Design of crRNA-hairpin-target RNA substrate for structural study. **e** Two views of the cryo-EM structure of the Cas13bt3-hpRNA complex. The domains are color coded according to domain structures in (**c**) and (**d**). **f** Two views of the atomic model of the Cas13bt3-hpRNA complex. Source data are provided in the Source Data File.

mismatched rAs or rUs at the end of the target (Fig. 2a). Mismatches at the 3′-end begin to affect the collateral cleavage when the duplex is equal to or shorter than 28 nt (T6), whereas a 25 bp or shorter (T9-T11) duplex activates Cas13bt3 to only around 20% of the activity with T1 (Fig. 2a, b). At the 5′-end of the target RNA, Cas13bt3 can tolerate up to 3 mismatches (Supplementary Fig. 1f, g). Our results confirm that a nearly 30 nt target is required to fully activate Cas13bt3[7], although 2-3 nt terminal mismatches can be well tolerated. Since a 25 bp spacer-target RNA duplex was used to assemble the complex and only a 20 bp duplex was observed in the previous ternary complex[27], the previously reported structure of Cas13bt3[27] might only represent an intermediate

state during activation (Cas13bt3[Int]), meaning additional conformational changes are required for Cas13bt3 to become fully active.

## Cryo-EM structure of activated Cas13bt3
To understand the structural basis of Cas13bt3 activation, we attempted to capture Cas13bt3 in a fully activated state with the catalytically inactive Cas13bt3 (dCas13bt3, R84A/H89A/R739A/H744A), crRNA and a 30 nt target. However, the dCas13bt3-crRNA-target ternary complex is unstable during cryo-EM sample freezing and no clear particles with the right size can be found on the cryo-EM grid. To stabilize the ternary complex, we designed a hairpin substrate by connecting the crRNA and

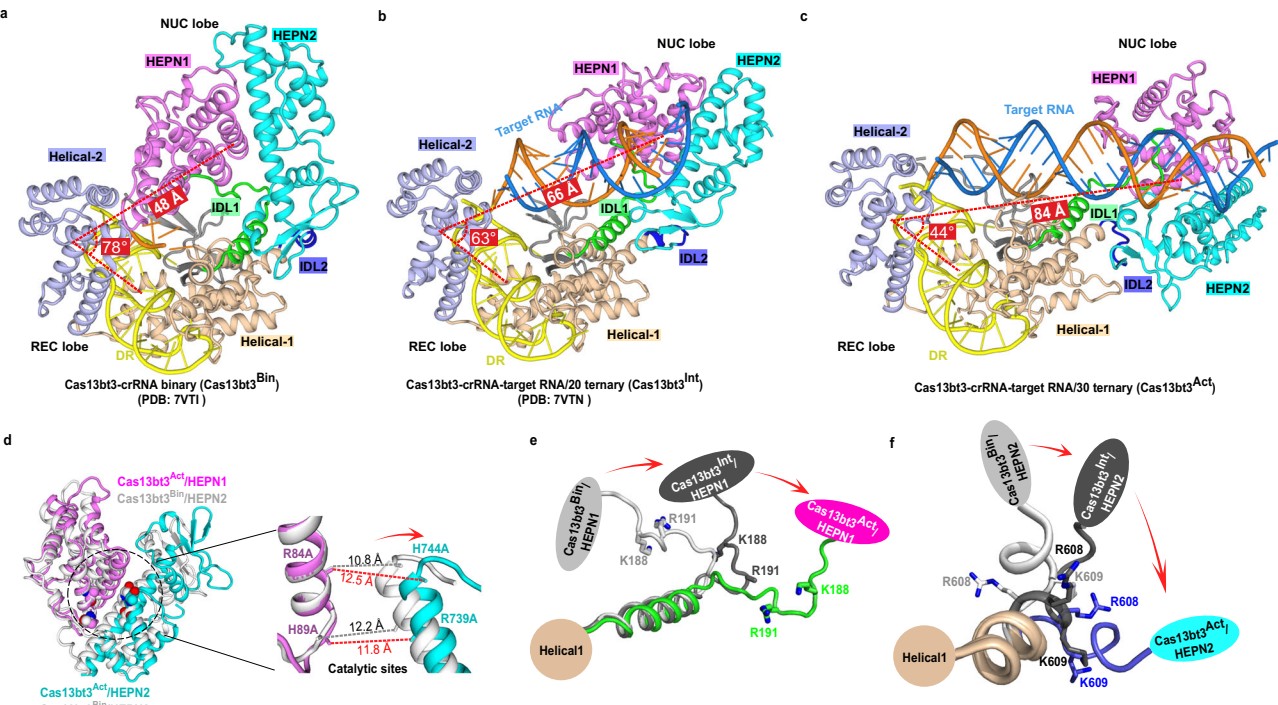

**Fig. 3 | Structural transformation during Cas13bt3 activation. a–c** Overall structures of the Cas13bt3[Bin] (**a**), Cas13bt3[Int] (**b**), and Cas13bt3[Act] (**c**). Large-scale conformational changes of HEPN domains are indicated by the angle between HEPN domains and DR and the distance between the HEPN domains and the DR. Color codes are defined as in Figs. 1c and d, respectively. **d** Overlay of the HEPN domains from Cas13bt[Bin] (gray) and Cas13bt[Act] (colored). The distance between the C-alpha atoms of R84 and H744, and H89 and R739 in the two structures are indicated, respectively. **e, f** Conformational changes of the IDL1 and IDL2 in Cas13bt3[Bin] (gray), Cas13bt3[Int] (black), and Cas13bt3[Act] (colored). The positively charged residues on IDL1 and IDL2 are highlighted.

target RNA with a 6 nt (CUUCUU) linker to stabilize the ternary complex (Fig. 2d, Supplementary Fig. 1h). The resulting crRNA-linker-target hairpin RNA (hpRNA) can activate Cas13bt3 collateral cleavage better than the unlinked crRNA-target RNA in the fluorescent assay (Supplementary Fig. 1c). Although possessing a low molecular weight of only 120 kDa, the dCas13bt3-hpRNA complex showed a good contrast on cryo-EM grids (Supplementary Fig. 2a). However, initial attempts at structural determination were hampered by the preferred orientation of the particles (Supplementary Fig. 2c, d). Combining data collected on Quantifoil and UltraGold grids and iterative 3D classification successfully identified a 3D class with high-resolution features and no orientation issue (Supplementary Fig. 2b, Supplementary Table 1). 3D auto refinement yielded a final map at 3.5 Å, with the core of the complex approaching 3 Å (Supplementary Fig. 2e–h). Most parts of the protein and the RNA can be traced and protein sidechains and nucleotide bases are well-ordered (Fig. 2c, e, Supplementary Fig. 3).

The structure of Cas13bt3-hpRNA ternary complex with the 30 bp duplex (from here on characterized as Cas13bt3[Act], the true active state according to our biochemical assays,) appears as a bilobed architecture with recognition (REC) and nuclease (NUC) lobes on two ends of the spacer/target duplex RNA (dsRNA) (Fig. 2e, f). A 36 nt direct repeat (DR) hairpin RNA of crRNA and a 30 bp spacer/target dsRNA were refined, with an angle of 45 degrees between the two (Fig. 2e, f). The REC lobe is located on the 5′-side of the target and consists of the Helical1, Helical2, and Lid domains for recognizing the DR (Fig. 2f, Supplementary FigsS. 3g, 4a, 5). Moreover, the Helical2 domain stacks at the end of the spacer/target dsRNA, whereas the Helical1 and Lid domains interact with the target near its 5′-end (Fig. 2e, f). The NUC lobe, comprised of the HEPN1 and HEPN2 domains, is on the distal end of the spacer/target dsRNA. In contrast to previous LbuCas13a[20] and EsCas13d[21] structures, there is no direct contact between the REC and NUC lobes in Cas13bt3 (Fig. 2f). Instead, the REC and NUC lobes are connected by two IDLs (inter-domain linkers): IDL1 (residues 183-215) is

between the HEPN1 and Helical1 domains, interacting with the spacer/target duplex; IDL2 (residues 607-614) locates near the HEPN active site and bridges Helical1 and HEPN2 domains (Fig. 2c, f).

Large-scale conformational changes between the REC and NUC lobes are evident when the Cas13bt3[Act] structure is compared to previous Cas13bt3-crRNA binary (Cas13bt3[Bin]) or Cas13bt3[Int] structures (Fig. 3a–c, Supplementary Fig. 4g–k). With the REC lobe aligned, the mass center of the NUC lobe in Cas13bt3[Act] moves ~40 Å and 20 Å and rotates 34 and 19 degrees relative to those in Cas13bt[Bin] and Cas13bt[Int], respectively (Fig. 3a–c, Supplementary Fig. 4c, d). Besides the N-terminal β-sheets in HEPN2 (Supplementary Fig. 4e, f), both REC and NUC lobes are similar to previous Cas13bt3[Bin] or Cas13bt3[Int] structures, with RMSDs around 2 Å (Fig. 3d, Supplementary Fig. 4b). Interestingly, the active site residue H744 from HEPN2 in Cas13bt3[Act] has moved 1.7 Å away from the R84 in the HEPN1 motif relative to the Cas13bt3[Bin] complex (Fig. 3d). The observed HEPN motif movement in Cas13bt3 is distinct from that in LbuCas13a and EsCas13d, where the active sites of both structures move closer upon target RNA binding to enable catalytic activity[20, 21] (Supplementary Fig. 6a–d). Besides the rigid-body movement of HEPN domains, there are drastic conformational changes in IDL1 and IDL2 (Fig. 3e, f). First, the IDL1 loop is dragged away from the REC lobe by HEPN1 movement to form new contacts with the spacer/target RNA duplex (discussed below). Second, the IDL2 loop undergoes large-scale structural transition relative to that in Cas13bt3[Bin] and Cas13bt3[Int] towards the HEPN active site, allowing the HEPN2 domain to move as a rigid body together with the HEPN1 domain. In addition, we compared our Cas13bt3 structure to the binary structures of PbuCas13b[23] or BzCas13b[22]. The REC lobe of Cas13bt3[Act] aligns well with the REC lobe in both PbuCas13 and BzCas13b, whereas the NUC lobes displayed large-scale conformational changes similarly as observed with respect to Cas13bt3[Bin] (Supplementary Fig. 6e, f). Moreover, the distance between two HEPN motifs in PbuCas13b binary complex[23] is similar to that in Cas13bt3[Act] (Supplementary Fig. 6g).

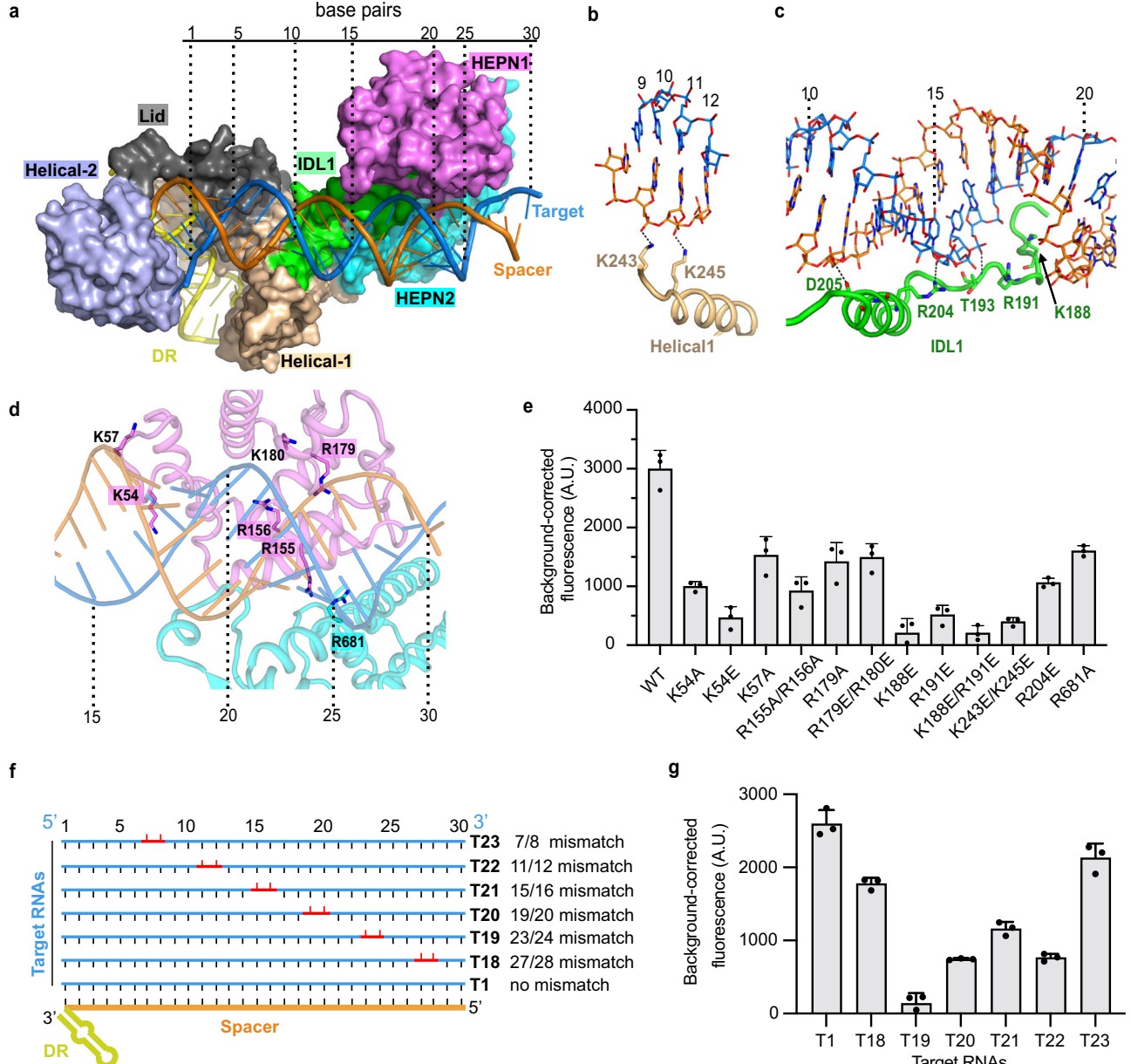

**Fig. 4 | Interaction of Cas13bt3 with RNA duplex. a** Spacer/target RNA duplex binding interfaces in Cas13bt3. The dsRNA are numbered according to the target RNA from 5′- to 3′-ends. **b-d** Zoom in views of interactions of the Helical1 loop (**b**), the IDL1 (**c**) and the HEPN domains (**d**) with dsRNA. Color codes are defined as in Figs. 1c and d, respectively. **e** Collateral activity of WT and mutant Cas13bt3 at the dsRNA binding interfaces. **f** Target RNAs design for examining effects of mismatch tolerance of Cas13bt3 cleavage. The mismatched RNA segments are colored in red. **g** Collateral activity of Cas13bt3 with mismatched bases at different positions along the target RNA. **e, g** R1-FQ was used as reporter RNA. Data are presented as means ± SDs (n = 3). A.U. relative fluorescence intensity in arbitrary units. Source data are provided in the Source Data File.

However, the HEPN motifs are further apart in BzCas13b binary structure[22] and additional conformational changes upon target binding are needed to bring them together (Supplementary Fig. 6h).

## Molecular basis of spacer/target RNA duplex recognition

The 30 bp spacer/target RNA duplex spans the REC and NUC lobes in Cas13bt3[Act]. However, direct protein-RNA contacts (distance < 3.5 Å) are scattered and only present in three areas, the Helical1 domain, the IDL1, and the HEPN1/HEPN2 domains (Supplementary Fig. 5). To easily track the protein-dsRNA interaction, we numbered the duplex according to the target sequence from 5′ to 3′ (Fig. 4a). At the 5′ side, a loop from Helical1 contacts the spacer strand and two positively charged residues K243 and K245 interact with the spacer at position 10-11 (Fig. 4b, Supplementary Fig. 5). A K243E/K245E mutation reduces

the Cas13bt3 activity by 86% (Fig. 4e). Notably, these salt bridges were not observed in Cas13bt3[Int], which is likely due to the 24-degree rotation of the spacer/target duplex relative to the crRNA (Supplementary Fig. 7e, f). At the 3′-end of the target, the two HEPN domains form a cleft for dsRNA binding (Fig. 4a, Supplementary Fig. 7b). Residues K54, K57, R179, K180, R155, and R156 from HEPN1 and R681 from HEPN2 are proximate to the RNA backbones (Fig. 4d). Consistent with the structural observation, mutations K54A, K54E, K57A, R155A/R156A, R179A and R179E/K180E from HEPN1 and R681A within HEPN2 displayed a reduction in activity by 66%, 85%, 49%, 70%, 52%, 50%, and 46%, respectively (Fig. 4e). Due to the relative tilting of the dsRNA to the HEPN domains, the positively charged dsRNA binding cleft is not utilized for dsRNA binding in Cas13bt3[Int] structure (Supplementary Fig. 7a–c). Instead, a different set of residues, including R122, R123,

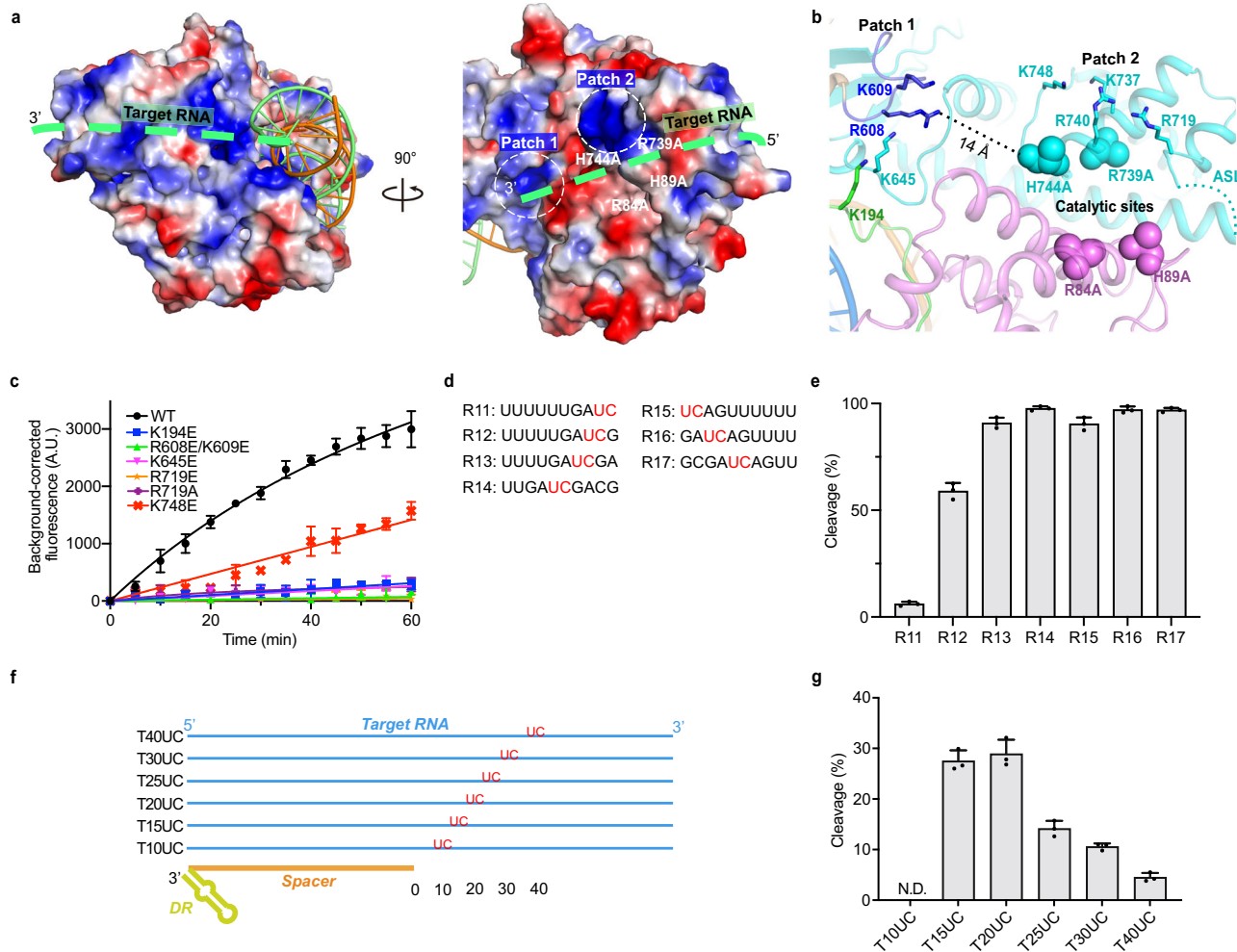

**Fig. 5 | Substrate binding surfaces of Cas13bt3. a** Two views of the charged potential surface of Cas13bt3. The Patch1 and Patch2 are circled. The possible path for target RNA is indicated by a green dotted line. **b** Zoom-in views of the Patch1 and Patch2 around the nuclease active site. The location of the active site is indicated by the catalytic residues R84A/H89A from HEPN1 and H744A/R739A from HEPN2 in sphere representation. The charged residues on Patch1 and Patch2 are shown in sticks. Color codes are defined as in Figs. 1c and d, respectively. **c** Collateral cleavage activities of WT and mutated Cas13bt3 on Patch1 and Patch2. R1-FQ was used as reporter RNA. Three replicates are performed and individual data points are shown. **d** Reporter RNA substrates design with the UC site at different positions. **e** Collateral cleavage activities of Cas13bt3 with substrates in (**d**). **f** Target RNAs design with UC site at different positions. **g** On-target RNA cleavage activities of Cas13bt3 with substrates in (**f**). 25 nM Cas13bt3-crRNA complex was used. **c**, **e**, **g** Data are presented as means ± SDs ($n = 3$). A.U. relative fluorescence intensity in arbitrary units. Source data are provided in the Source Data File.

R155, R156, K169, and K170 in the HEPN1 domain, and K645 in the HEPN2 domain, are proximate to the dsRNA in Cas13bt3[Int] (Supplementary Fig. 7d). Moreover, the fully stretched IDL1 in Cas13bt3[Act] interacts with the spacer/target RNA duplex near position 11-20 (Fig. 4c). Residue K188, R191 R204, D205 and the backbone of T193 are near the major groove of the spacer/target duplex (Fig. 4c, Supplementary Fig. 5). The mutations R204E, K188E, R191E, and K188E/R191E dramatically reduce activity by 65%, 92%, 83%, and 93%, respectively (Fig. 4e). The IDL1 interactions are missing in the Cas13bt3[Int] due to the insufficient movement of the HEPN domains (Supplementary Fig. 7g).

The scattered protein-RNA interactions in Cas13bt3[Act] structure implicated that the Cas13bt3 may tolerate mismatches in target RNA (Fig. 4a). Using target RNA with double mismatches along the duplex (Fig. 4f), we showed that mismatches at positions 7−8 and 27−28 have a minor effect on Cas13bt3 (Fig. 4g). In contrast, mismatches at positions 11/12, 15/16, 19/20, and 23/24, which are involved in protein RNA interactions in our Cas13bt3[Act] complex, reduce Cas13bt3 activity by 70%, 55%, 71%, and 94%, respectively (Fig. 4g). Our biochemical assay is also consistent with the mismatch tolerance result in Cas13bt3-induced knockdown assays in vivo[7].

## Structural basis of Cas13bt3 activation

Considering the limited conformational changes between the catalytic HEPN motifs (Fig. 3d), we suspect additional allosteric changes near the active site may contribute to HEPN activation. We identified two positively charged patches around the active sites that went through conformational transitions upon long target RNA binding (Fig. 5a, b). Patch1 is composed of sidechains of R608 and K609 from IDL2, K645 from HEPN2, and K194 from IDL1 (Fig. 5b). Patch1 was only formed upon the long target RNA binding, and the R608, K609, K194, and K645 in Patch1 translocated 5.6, 4.7, 11 and 15 Å compared to corresponding residues in Cas13bt3[Int] (Supplementary Fig. 7h-l). Mutations at K194E, R608E/K609E, and K645E almost eliminate the HEPN cleavage (Fig. 5c), even though the target can still bind to mutated Cas13bt3 with a similar affinity as wild-type Cas13bt3 (WTCas13bt3) (Supplementary Fig. 8). As the patch is over 10 Å away from the active site, we envision at least 2–3 nt is needed to span the space (Fig. 5b). Consistent with our structural observations, two nucleotide overhangs are required at the 3′ side of the UC motif for optimal cleavage, whereas the overhanging nucleotides are not strictly needed at the 5′-end (Fig. 5d, e, Supplementary Fig. 9a).

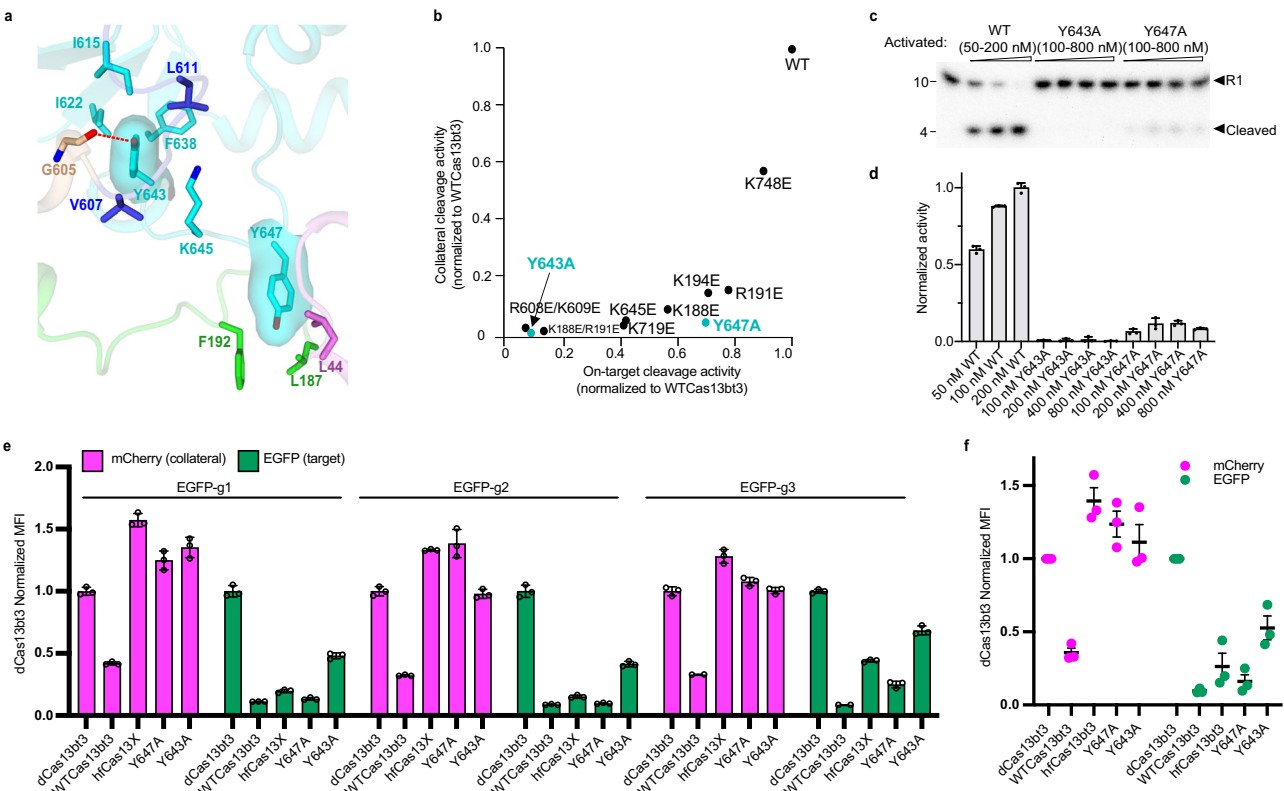

**Fig. 6 | Engineering of Cas13bt3 for high-fidelity cleavage. a** Two hydrophobic clusters around residue K645. Color codes are defined as in Fig. 1c. **b** Normalized on-target and collateral cleavage activities of mutated Cas13bt3. **c** Representative gel image of the collateral cleavage of WTCas13bt3, Y643A, and Y647A at different protein concentrations. RNA sizes (in nt) are indicated. **b**, **c** At least three times each experiment was repeated independently with similar results. **d** Normalized cleavage activity as shown in (**c**). **e** On-target and collateral cleavage of Cas13bt3 variants in HEK293T cells. Differential changes of normalized MFI were induced by dCas13bt3, WTCas13bt3, hfCas13bt3, Y647A, and Y643A in HEK293 cell lines. **f** Statistical data analysis for relative degradation efficiency by Cas13bt3 variants from (**e**). Normalized MFI, mean fluorescence intensity relative to the dCas13bt3 condition. **d**–**f** Data are presented as means ± SDs (n = 3). Source data are provided in the Source Data File.

Patch2 locates above the active sites and is composed of the sidechains of R719, K737, R740, and K748 from HEPN2 (Fig. 5a, b, Supplementary Fig. 9c). R719 sits on the active site loop (ASL, residues 703–729), a motif that was proposed to block RNA substrate entrance in the absence of target[27]. ASL is ordered in Cas13bt3[Bin], disordered in Cas13bt3[Int], and becomes partially ordered in Cas13bt3[Act], with residues 717–729 taking a similar conformation as in the Cas13bt3[Bin] (Supplementary Fig. 4f). Mutation of this residue to R719A or R719E reduces the activity by 94% (Fig. 5c), indicating that R719 may stabilize the substrate binding for HEPN cleavage, instead of active site blockage. Mutation of K748E on Patch2, which is further away from the active site than R719, reduces Cas13bt3 cleavage by 50% (Fig. 5c).

In addition to Patch1 and Patch2, we could identify continuous positively charged surfaces from the 3′-end of the target to the active site, suggesting a path for the target RNA cleavage (Fig. 5a). When the UC site is placed 10 nt away from the end of the duplex, no target cleavage can be detected. Target cleavage activity is optimal when the UC site is positioned 15–20 nt away from the end of the duplex, but the UC site at 30–40 nt away from the duplex end reduces the activity (Fig. 5g, Supplementary Fig. 9b). The observed on-target cleavage pattern is consistent with the positively charged path on the Cas13bt3 surface for target binding and cleavage (Fig. 5a).

### Structural-guided engineering of Cas13bt3

For in vivo transcriptional modulation and RNA knockdown, a Cas13 system possessing efficient target cleavage while maintaining minimum non-specific cleavage is required to reduce the transcriptome damage at targeted cells. To engineer Cas13bt3 with

minimal non-specific activity, we tested a series of mutations on the IDLs and the positively charged Patches for in vitro cleavage (Supplementary Fig. 9e–h). We found that the K188E/R191E double mutation on IDL1 almost abolished both cleavage activities (Fig. 6b, Supplementary Fig. 9e–h), while K188E and R191E single mutation keeps 55% and 77% on-target cleavage, and retains about 10% and 15% collateral cleavage activity, respectively (Fig. 6b, Supplementary Fig. 9e–h). The double mutations of R608E/K609E on Patch1 showed dramatically reduced cleavage activity of both the target and collateral cleavage, while mutations of K194E and K645E on Patch1 and R719E on Patch2 keep 70%, 41%, and 40% on-target cleavage activity but only retains 14%, 5%, 2% collateral cleavage activity, respectively (Fig. 6b, Supplementary Fig. 9e–h). These results suggested that the positively charged patches near the active site may contribute more to non-specific RNA binding contributing to collateral cleavage events as compared to the target RNA that can base-pair with the crRNA.

As revealed through random mutagenesis screening[29], perturbing hydrophobic interactions near the positively charged patches may affect local folding and RNA binding. The mutations of K645E on Patch1 or R719E on Patch2 dramatically decrease collateral cleavage activity to less than 10% of WTCas13bt3 but keep about 40% on-target activity, which indicated two hotspots for further engineering. Based on our structure, R719 is on the flexible ASL[27], while two hydrophobic clusters can be found near K645, one around Y643 and the other near Y647 (Fig. 6a). Mutation of Y643A almost abolished both cleavage activities, while Y647A significantly reduces collateral cleavage to 4% but still keeps 70% activity of on-target cleavage (Fig. 6b, Supplementary Fig. 9e, f). Furthermore, we measured the collateral cleavage

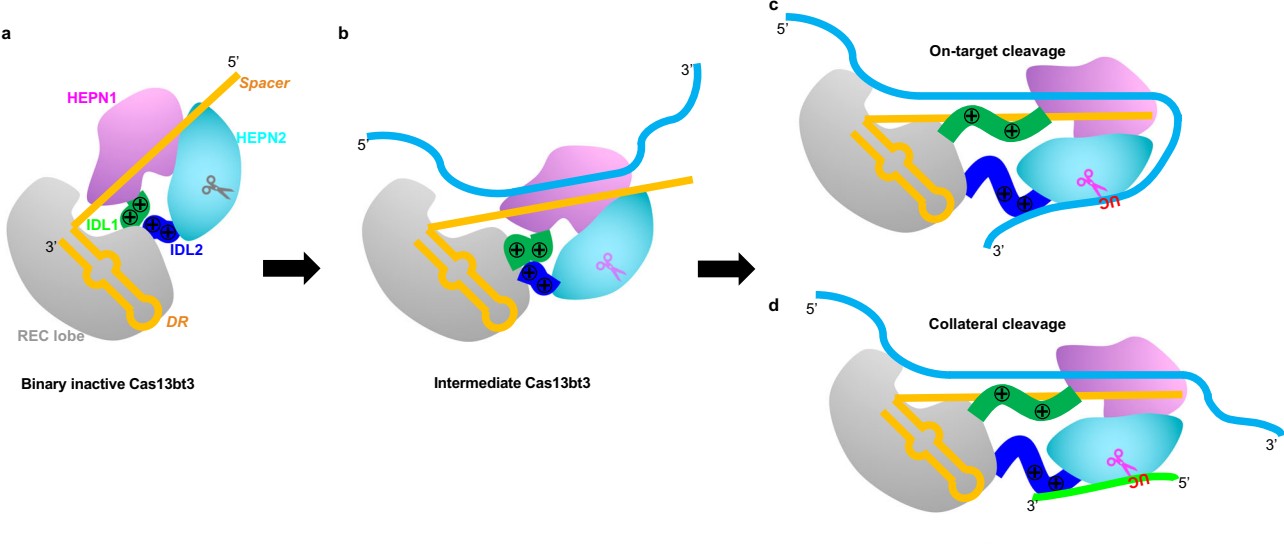

**Fig. 7 | Model for Cas13bt3 catalyzed crRNA-guided RNA cleavage. a** The inactive state of the Cas13bt3-crRNA binary complex, HEPN domains are near the REC lobe. **b** The intermediate state of Cas13bt3-crRNA complexed with a 25 nt target RNA. HEPN domains undergo rigid-body movement while IDL1 and IDL2 are not fully stretched. **c**, **d** The active state of Cas13bt3-crRNA complexed with a 30 nt target RNA. HEPN domains migrate further down the target-spacer duplex, IDL1 becomes fully stretched to interact with spacer/target duplex, and IDL2 forms binding surface for RNA cleavage. Color codes are defined as in Fig. 1c and d, respectively. The UC cleavage site on target RNA or reporter RNA are indicated and colored in red.

at increasing protein concentrations using WTCas13bt3 as the control. WTCas13bt3 fully cleaved the substrate at a concentration of 200 nM, Y647A only keeps 10% activity with a concentration at up to 800 nM, and Y643A shows almost no collateral activity at any concentration (Fig. 6c, d). These results further verify the critical role of Patch 1 for Cas13bt3 activation.

To evaluate the collateral effects of the structurally informed mutants (Y643A or Y647A) in mammalian cells, we took advantage of a well-established dual-fluorescence approach based on the plasmid (Addgene #190033) which contains the previously engineered high-fidelity Cas13X (hfCas13X or hfCas13bt3)[29], together with EGFP, mCherry and an EGFP-targeting guide RNA in one plasmid. The hfCas13bt3 was developed from random mutagenesis screening and contains double mutations Y672A/Y676A (Supplementary Fig. 9d). Based on the plasmid backbone of hfCas13bt3, we designed three guide RNAs that target EGFP and constructed the WTCas13bt3, dCas13bt3, as well as the mutant containing either Y643A or Y647A for each guide RNA (see methods). Using flow cytometry (Supplementary Fig. 10), we found that WTCas13bt3 transfection induced significant reductions in not only EGFP fluorescence (caused by on-target degradation) but also mCherry fluorescence (caused by collateral degradation) compared with dCas13bt3 (Fig. 6e, f). The mutant Y647A exhibited a similar level of on-target cleavage activity as WTCas13bt3 with guide RNA 1 and 2 and slightly lower activity with guide RNA 3. Moreover, the degradation of targeted transcripts by Y647A is better than hfCas13bt3 across all three guide RNAs (Fig. 6e, f). Interestingly, the plasmid containing hfCas13bt3, Y643A, or Y647A exhibited relatively similar percentages of mCherry fluorescence compared with dCas13bt3 (Fig. 6e, f), consistent with their low collateral activities. The cellular assays confirm that our rationally designed Cas13bt3 Y647A, possessing minimal collateral effects while maintaining robust on-target cleavage activity, is a promising tool for in vivo RNA editing and further therapeutic applications.

## Discussion

Our structural and biochemical characterizations of Cas13bt3 suggest unique characteristics of target RNA recognition and activation

mechanism (Fig. 7). Although with a similar bilobed architecture as other Cas13s, the two HEPN domains in the NUC lobe of Cas13bt3 are at two ends of the polypeptide chain and only connect to the REC lobe with two IDLs. As shown in biochemical assays and structures, Cas13bt3 undergoes target length-dependent activation and gradual HEPN domain movement. It is likely that the target RNA binds initially to the spacer with its central region as a seed, then threads to the distal ends. Only when the 30 nt target RNA binds and is fully complemental to the spacer does the IDL1 domain become fully stretched for target RNA binding. Meanwhile, the IDL2 migrates close to the active site and forms a new interface for supporting HEPN nuclease cleavage. This model of Cas13bt3 target recognition and activation is well-suited for its small size and limited RNA interfaces but distinct from Cas13a[20] and Cas13d[21], where the dsRNA binding site is pre-formed and only enlarged by interdomain rotations upon target binding (Supplementary Fig. 6a–d). Similar target length-dependent activation has been observed in RNA-guided DNA nuclease Cas12i, where Cas12i undergoes a two-step activation by the propagation of the crRNA-DNA hetero-duplex coupled with the dynamic conformational change of a lid motif[30]. Recent bioinformatic analysis identified a large number of new Cas13s with similar domain structures as Cas13bt3[26], where the HEPN domains are at the two ends of the polypeptide chain and connected to the REC domains with IDLs (Supplementary Fig. 11). These new Cas13 effectors may take a similar activation mechanism as Cas13bt3.

Modulating collateral cleavage remains an essential hurdle to utilizing Cas13 for various applications. Our biochemical data suggested that the positively charged surfaces around the HEPN active site are important for Cas13 activity, which is possibly due to reduced target RNA binding and pairing, reporter/target RNA binding for HEPN cleavage, or catalysis itself. The EMSA binding assays suggest that the Patch 1 and Patch 2 mutations have minimal effect on target RNA binding (Supplementary Fig. 8). Their locations and the charge-reducing/reversal nature of the mutations indicate that these mutations are most likely to affect Cas13bt3 activities by reducing the reporter/target RNA binding for HEPN nuclease cleavage. Considering the extensive interaction of the target RNA with the spacer and Cas13, alterations of the charged surfaces near the active site may have a bigger impact on

the non-specific RNA cleavage than on-target cleavage. The Y647A or Y643A mutant around Patch1 may not affect RNA substrate binding but could disrupt the local folding or the orientation of positively charged residues on Patch 1, which result in reduced activity. As predicted, our mutagenesis of Cas13bt3 suggested that residues near the positive Patch1 reduce collateral cleavage but retain most of the on-target activity in both biochemical assays and mammalian cells. On the other hand, collateral cleavage of Cas13 has been leveraged for ultra-sensitive RNA detection in vitro. Increasing the non-specific RNA binding surface by linking an RNA-binding domain to Cas13a has been shown to significantly increase collateral RNA cleavage activity and RNA detection sensitivity[31]. Thus, structural-guided perturbations of the substrate binding surfaces of Cas13 could be an effective way for modulating Cas13 activities for various applications.

## Methods

### Protein cloning, expression, and purification

The gene of dCas13bt3 with R84A/H89A/R739A/H744A mutations in the HEPN domains, was synthesized by GeneUniversal and cloned into a modified pET-SUMO vector (Novage) with an N-terminal His6-SUMO tag following a Sentrin/small ubiquitin-like modifier (SUMO)-specific protease 2 (SEPN2) cleavage site. The WTCas13bt3 was constructed by a In-Fusion Snap Assembly method (TaKaRa), while other Cas13bt3 mutations were constructed using the WTCas13bt3 as the template. The expression plasmids for WT and mutated Cas13bt3 will be available upon request.

To test the collateral effects of Ca13bt3 mutants in mammalian cells, we generated the plasmids based on the template of hfCas13bt3 (Addgene #190033). First, the spacer sequence of guide RNA 1 (GTCCTCCTTGAAGTCGATGCCCTTCAGCTC), 2 (AGCACTG CACGCC GTAGGTCAGGGTGGTCA), or 3 (GCAGGACCATGTGATCGCGCTTCTC GTTGG) was generated by ligating phosphorylated and annealed oligos (IDT) into the corresponding BbsI-digested backbone of hfCas13bt3, respectively. Then, the dCas13bt3, and mutant containing Y643A or Y647A were constructed by the In-Fusion Snap Assembly method based on the plasmid of Cas13bt3.1 (Addgene #171379), and the gene sequence of WTCas13bt3, dCas13bt3, and mutant containing Y643A or Y647A was respectively ligated to AgeI and HindIII-double digested backbones of the spacer-modified hfCas13bt3 plasmid. All the constructions were confirmed by sanger sequencing (Genewiz). The DNA primers for cloning were listed in Supplementary Table 2.

Expression and purification of Cas13bt3 were performed as previously described with modifications[25]. Briefly, recombinant Cas13bt3 protein was overexpressed in Escherichia coli Rosetta (DE3) in Terrific Broth (TB) medium. The cells were grown at 37 °C until OD600 reached 0.8, then the cells were cooled on ice for 20 min and switched to 18 °C and induced with 0.2 mM isopropyl b-D-1-thiogalac- topyranoside (IPTG) for 16–18 h. Cell pellets were harvested with centrifugation at $5000 \times g$ at 4 °C, and resuspended in buffer A (40 mM Tris-HCl, pH 8.0, 500 mM NaCl, 5 mM b-mercaptoethanol, 1 mM phenylmethylsulfonyl fluoride), and lysed by sonication, then centrifuged at $23,000 \times g$ for 1 h at 4 °C. The supernatant was passed through 2 ml Nickel-NTA resin (Qiagen), and the resin was washed with 20 column volumes of lysis buffer, then the protein was eluted from the column with buffer B (40 mM Tris-HCl, pH 8.0, 500 mM NaCl, 5 mM b-mercaptoethanol, 1 mM phenylmethylsulfonyl fluoride, 250 mM imidazole). The SUMO tag was cleaved overnight on ice by SEPN2 protease (homemade). After cleavage, the protein was loaded onto a 5 ml heparin column (GE Healthcare) which was preequilibrated with the buffer C (20 mM Tris-HCl, pH 8.0, 500 mM NaCl, 1 mM DTT). Protein was eluted by a gradient NaCl concentration with buffer D (20 mM Tris-HCl, pH 8.0, 1 M NaCl, 1 mM DTT). Peak fractions were pooled, concentrated, and supplemented with 30% glycerol. The purified Cas13bt3 protein was aliquoted and flash-frozen at −80 °C for further study. The mutants were purified by the same method as described above.

### In vitro transcription (IVT) and purification of RNAs

The 120 bp transcription template for hpRNA containing 17 bp T7 RNA polymerase promoter and 103 bp transcription template was ordered as 8 DNA oligos (hpIVT-S1 to hpIVT-S4, IDT). The S1 to S4 were phosphorylated with T4 polynucleotide kinase (10 units/µl; Thermo Scientific), annealed and ligated into the pUC19 vector (Thermo Fisher) with HindIII and XbaI restriction sites. The construction was confirmed by sanger sequencing (Genewiz), then the transcription template was obtained by PCR amplification with primers of M13F and hpIVT-R. The PCR product was purified with phenol and followed by 95% ethanol precipitation, then the product was dried and dissolved with RNase-free $H_2O$. The transcription templates of crRNA (crIVT) and short target RNAs (T1IVT-T23IVT) were ordered as oligonucleotides from IDT, and the double-stranded transcription template for each RNA was obtained by annealing the forward primer and the reverse primer at a molar ratio of 1:1. The constructions containing transcription template of long target RNAs (T0, T10UC, T15UC, T20UC, T25UC, T30UC and T40UC) were prepared by the same method as hpRNA, and each transcription template was obtained by PCR amplification with M13F and T0-R or TUC-R, respectively. The DNA oligos for preparing RNA transcription templates and corresponding RNA sequences were listed in Supplementary Table 3 and Table 4.

The in vitro transcription was carried out at 37 °C for 4 h in buffer containing 100 mM HEPES potassium (pH7.5), 30 mM DTT, with 20 mm $MgCl_2$ for crRNA, hpRNA and long target RNAs or 8 mM $MgCl_2$ for short target RNAs, 2 mM spermidine, 2.5 mM each NTP, 100 ng/µl DNA template and 2 µM T7 RNA polymerase (homemade). The RNAs were purified by 12% denaturing (7 M urea) polyacrylamide gel electrophoresis (PAGE), extracted, and precipitated by ethanol. The RNAs pellet was dissolved with RNase-free $H_2O$, and the concentration were measured with Nanodrop (DeNovix DS-11). The crRNAs and hpRNA were diluted to 10 µM and annealed by heating to 75 °C for 5 min, then slowly cooling to room temperature, then aliquoted and flash-frozen at −80 °C for further use.

For 5′-end radiolabeling of target RNAs, the phosphorylated 5′ end of target RNAs from IVT was removed by phosphatase (NEB) and purified with Illustra MicroSpin G-50 columns (GE Healthcare), then 2 µM of de-phosphorylated target RNA was incubated in a 20 µl reaction mixture containing 1 X T4 polynucleotide kinase buffer, 0.5 µl of [γ-$^{32}$P] ATP (6,000 Ci/mmol; PerkinElmer Life Sciences), and 1 µl of T4 polynucleotide kinase (10 units/µl; Thermo Scientific). The reaction was incubated at 37 °C for 30 min. The labeled product was purified with Illustra MicroSpin G-50 columns. For 5′-end radiolabeling of ssRNA (R1-R17), the RNA oligos were ordered from IDT without phosphorylated 5′ end and were directly subjected to radiolabeling by the same method as described above. The concentration of 5′-end radiolabeled RNAs was measured with Nanodrop. The eluted RNA was aliquoted and stored at −20 °C if not immediately used. The 5′ fluorophore (FAM) labeled target RNA T1 (5′-FAM-T1), U20 (5′-FAM-U20), ssRNA R1 (5′-FAM-R1), and ssRNA36 (5′-FAM-ssRNA36) or 5′-fluorophore and 3′-quencher labeled R1 (R1-FQ) were ordered from IDT.

### Cryo-EM sample preparation

The complex of Cas13bt3-hpRNA was assembled by mixing the purified dCas13bt3 and the 103 nt hpRNA at a molar ratio of 1:2 in buffer containing 20 mM HEPES, pH 7.5, 400 mM NaCl, 5 mM $MgCl_2$, and 1 mM DTT. The complex was purified through two steps of gel filtration chromatography, first at high salt to purify the Cas13bt3 complex with specific hpRNA binding and second at low salt to reduce the salt concentration for stabilizing the ionic interactions between Cas13bt3 and hpRNA (Supplementary Fig. 1h). First, the complex was incubated on ice for 30 min and purified with Superdex 200 Increase 10/300 column in buffer (20 mM HEPES, pH 7.5, 400 mM NaCl, 5 mM $MgCl_2$, and 1 mM DTT). The peak fractions of the complex were collected, concentrated, and the buffer was exchanged to the buffer containing

20 mM Tris-HCl, pH 8.0, 100 mM NaCl, 5 mM $MgCl_2$, and 1 mM DTT with an Amicon Ultra centrifugal filter unit with 50 kDa cutoff (Millipore). Second, the above 500 µl concentrated complex was further purified with Superdex 200 Increase 10/300 column in buffer (20 mM HEPES, pH 7.5, 100 mM NaCl, 5 mM $MgCl_2$, and 1 mM DTT). The peak fractions of the complex were collected and concentrated to 0.9 mg/ml, as determined by the Bradford assay kit (Bio-Rad).

3 µl freshly purified Cas13bt3_hpRNA complex were spotted onto freshly glow-discharged Quantifoil R 1.2/1.3 Cu 300 mesh grids (Quantifoil) or GF 1.2/1.3 AU 300 mesh grids (Protochips). Excess samples were blotted using the Vitrobot Mark IV (FEI) with the standard Vitrobot filter paper (Ø55/20 mm (Ted Pella), the blotting time was set to 2 s, the blotting force was set to 5 and the blotting was done under 100% humidity at 20 °C. The grids were flash-frozen in liquid ethane and stored in liquid nitrogen.

### Cryo-EM data collection and processing

The images were recorded on a Titan Krios electron microscope operated at 300 kV (cryo-EM core facility at University of Texas McGovern Medical School) using the super-resolution mode with a nominal magnification of 165 K (calibrated pixel size of 0.87 Å on the sample level, corresponding to 0.435 Å in super-resolution mode). Movies were recorded with a K2 Summit camera, with a dose rate of 7 e-/pixel/s with a total exposure time of 7 s, with 0.2 s for each frame, generating 35 frames per micrograph. Defocus values range was set between 0.6 and 2.2 µM. For dataset 1, 2236 micrographs were collected from Quantifoil R 1.2/1.3 Cu 300 mesh grids. For dataset 2, 3358 micrographs were collected from GF 1.2/1.3 AU 300 mesh grids. Micrographs from datasets 1 and 2 were motion corrected by MotionCor2[32], and defocus values were estimated on non-dose-weighted micrographs with Gctf[33], respectively.

About 110 K particles were picked from 100 manually screened micrographs from dataset 1 based on reference-free auto-picking (Laplacian-of-Gaussian picking) through RELION-4.0[34,35]. The particles were extracted to a pixel size of 5.2 Å and were subjected to 2D classification, and good classes were selected and used as templates for reference-base picking of both datasets 1 and 2. Around 4,798,775 particles from dataset 1 and 7,511,757 particles from dataset 2 were picked. The picked particles were extracted to a pixel size of 5.2 Å and around 1,735,702 particles from dataset 1 and 918385 particles from dataset 2 were selected after 2D classification, respectively. The selected particles were extracted (0.87 Å/pixel), combined, and imported to cryoSPARC-3.3[36] for another round 2D classification, and about 1,760,928 particles were selected. The selected particles were separated into six classes by cryoSPARC ab-initial reconstruction, setting the maximum resolution to 8 Å, and one class with intact and clear protein and RNA features was selected for subsequent processing. To solve the preferred orientation issue, iterative ab-initial reconstructions were performed three more times by setting the maximum reconstruction resolution to 6 Å and separating into three classes each time, during each round of 3D classification only the class with clear protein and RNA features was selected. About 37228 particles were selected from cryoSPARC and were imported and subjected to CTF refinement, and Bayesian polishing in RELION-4.0. The particles after polishing were imported to cryoSPARC-3.3, and final refinements were done with the non-uniform refinement[37], which yielded a map for subsequent modeling with global resolutions of 3.52 Å according to the Fourier shell correlation (FSC) = 0.143 criterion[38]. The local resolution was estimated by cryoSPARC-3.3. Additional processing details are summarized in Supplementary Fig. 2.

### Model building and refinement

A homolog model of Cas13bt3 was generated with the AlphaFold Colab server (https://colab.research.google.com/github/deepmind/alphafold/blob/main/notebooks/AlphaFold.ipynb). The model of DR was adapted from PbuCas13b-crRNA binary structure (6DTD)[23], and the model of double-stranded RNA formed by the spacer and target RNA was generated from Coot-0.9.6 modeling tools[39]. Each domain of Cas13bt3, and RNAs were manually docked into the cryo-EM density maps in Chimera[40]. The model was further manually rebuilt in COOT based on electron density, and refined in Phenix with real-space refinement and secondary structure and geometry restraints[41]. No clear electron density was observed for 6 nucleotides linker between the spacer and target RNA, poor electron density was observed for the residues 227–229 of the Helical1 domain and 272–278 of the Lid domain. All figures were generated by UCSF Chimera and PyMol (http://www.pymol.org). Statistics of all cryo-EM data collection and structure refinement are shown in Supplementary Table 1.

### In vitro RNA cleavage gel assay

Both target RNA cleavage and collateral cleavage reactions were performed in a cleavage buffer containing 20 mM Tris.HCl, pH 8.0, 60 mM NaCl, 2 mM $MgCl_2$, 1 mM DTT, 5% glycerol. For target RNA cleavage assays, 1 µM of Cas13bt3 and crRNA were incubated at a molar ratio of 1:1 on ice for 10 min in the cleavage buffer, then the reactions were initiated by mixing 50 nM or indicated concentration of Cas13bt3-crRNA complex with 100 nM indicated target RNA in the cleavage buffer. For collateral cleavage assays, 1 µM of Cas13bt3, crRNA, and target RNA (T1) were incubated at a molar ratio of 1:1:1 on ice for 10 min in the cleavage buffer, then the reactions were initiated by mixing 50 nM or indicated concentration of Cas13bt3-crRNA-target complex (activated Cas13bt3) with 100 nM indicated reporter RNA. All reactions were performed at 37 °C for 30 min, and terminated by adding 2× loading buffer (93.5% formamide, 0.025% xylene cyanol FF, and 50 mM EDTA, pH 8.0) and quenched at 85 °C for 5 min. Quenched reactions were resolved on 12% denaturing polyacrylamide gels (Urea-PAGE) for target RNA cleavage assays, or 20% Urea-PAGE for collateral RNA cleavage assays. For the assays using 5′-end radiolabeled RNAs as substrates, the gels were followed by exposure to a phosphorimager plate for 1 h, and imaged using phosphorimaging by the Sapphire Biomolecular Imager (Azure Biosystems). For the assays using 5′-FAM-labeled RNAs as substrates, the gels were directly visualized and captured with a Sapphire Biomolecular Imager (Azure Biosystems). Assays were performed in at least three independent replicates, and the band intensity of the substrates and products were analyzed using Azure Spot (Azure Biosystem) and plotted through GraphPad Prism.

For Supplementary Fig. 1b, to map the cleavage sites on ssRNA36, the RNA size controls (M1 and M2) were made by digesting the 5′-FAM-ssRNA36 with RNase T1 (Thermo Scientific, catalog EN0541) or Exonuclease T (RNase T, NEB, catalog M0265S), respectively. RNase T1 is an endoribonuclease that specifically degrades single-stranded RNA at G residues. RNase T is a single-stranded RNA or DNA-specific nuclease that removes nucleotides in the 3′ to 5′ direction, but a single 3′-terminal C residue can reduce RNase T action by 100-fold and two consecutive terminal C residues will stop the enzyme.

### Fluorescence plate reader assay

For Supplementary Fig. 1c, to compare the collateral activity of WTCas13bt3 activated by crRNA and target RNA T1 or by hpRNA, WTCas13bt3, crRNA, and T1 or WTCas13bt3 and hpRNA were incubated at a molar ratio of 1:1:1 or 1:1 on ice for 10 min in the cleavage buffer, then 10-µl cleavage reactions were initiated by mixing the indicated concentration of WTCas13bt3-crRNA-T1 complex or WTCas13bt3-hpRNA complex with 100 nM R1-FQ in cleavage buffer.

For testing the collateral activity of Cas13bt3 mutants, 1 µM of WTCas13bt3 or mutants and crRNA were incubated at a molar ratio of 1:1 on ice for 10 min in the cleavage buffer first, then 10-µl cleavage reactions were initiated by mixing 50 nM protein-crRNA complex with 100 nM target RNA T1, and 100 nM R1-FQ in cleavage buffer. All reactions were performed in a 384-well microplate (Greiner, 784900) at

37 °C, with fluorescence monitored (excitation: 490 nm, emission: 520 nm) every 5 min for 60 min on a TECAN infinity M200 plate reader, and data were normalized to the first-time point.

## Electrophoretic mobility shift assay

To quantify the target RNA binding affinity, assays were carried out in a binding buffer containing 20 mM Tris.HCl, pH 8.0, 60 mM NaCl, 2 mM $MgCl_2$, 1 mM DTT, 5% glycerol. WTCas13bt3 and mutants were respectively complexed with crRNA in binding buffer at a molar ratio of 1:1 for 15 min on the ice, then a serial dilution of protein-crRNA complex (from 50 to 300 nM) was incubated with 10 nM 5′-FAM-T1 in binding buffer for 30 min on ice, respectively. Then each sample was mixed with a 5× loading buffer containing 10% glycerol and 0.05% bromophenol blue before loading onto a 5% native polyacrylamide gel containing 0.5× TBE buffer. The gel was pre-run at 4 °C for 45 min at 120 V with 0.5× TBE as a running buffer. Gels were imaged using phosphorimaging by the Sapphire Biomolecular Imager (Azure Biosystems). The bound and unbound fraction of target RNA was quantified by using Azure Spot (Azure Biosystem), plotted in GraphPad Prism, and fitted by specific binding with Hill slope.

## Cell Culture, transfection, and flow cytometry

HEK293T cells (American Type Culture Collection, catalog CRL-3216) were cultured in 10 cm treated, vented plates (Greiner Bio-One). The culturing media was Dulbecco's modified Eagle's Medium (DMEM) plus GlutaMAX (Gibco) which contained fetal bovine serum at 10% v/v (Gibco) and Penicillin-Streptomycin 1% v/v (Gibco). Cells were passaged between 80-90% confluency. The cell status used in all experiments was between Passage 1 to Passage 10. They were grown in an incubator set at 37 °C and 5% $CO_2$.

To prepare for an experiment, cells were seeded at a concentration of $2 \times 10^5$ cells in 500 µl of DMEM, with cell vitality above 95% and plate confluency around 90%. Cell vitality was assayed through Typhan Blue (Invitrogen) staining, placed on a Countess Cell Counting Chamber Slide (Invitrogen) and counted using a Countess II FL Automated Cell Counter (Thermo Fisher Scientific), with the average of the two concentration and vitality measurements used. After seeding, cells were left at room temperature for 15 min to mitigate evaporation.

After 12 h, cells were transfected. For each well, 800 ng of plasmids were buffered to a final volume of 25 µl in Opti-MEM I Reduced Serum Medium (OMEM) (Thermo Fisher Scientific). Along with the DNA, 2.5 µg of Polyethyleneimine (PEI) (Thermo Fisher Scientific) was buffered to a final volume of 25 µl in OMEM. The two were mixed and allowed to incubate at room temperature for 10 min before the total 50 µl was transfected into each well.

Forty-eight hours later, cells were analyzed for mCherry fluorescence and EGFP fluorescence by flow cytometry using a SONY SA3800. Single stain controls were predetermined for all experiments by using HEK293T cells transfected with either a single mCherry reporter or a single EGFP reporter. All flow cytometry data were analyzed using FlowJo v10.4.0.

## Reporting summary

Further information on research design is available in the Nature Portfolio Reporting Summary linked to this article.

## Data availability

The three-dimensional cryo-EM density maps for Cas13bt3$^{Act}$ complexes have been deposited in the EM Database under the accession code EMDB: EMD-29433, and the coordinates for the structure have been deposited in Protein Data Bank under accession code PDB 8FTI. The raw micrographs for the cryo-EM data will be available upon request. The rest of the data presented are available in the article, Supplementary Information and Source Data File. Source data are provided with this paper.

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

## Acknowledgements

We acknowledge Welch Foundation (C-2033-20200401) and the Cancer Prevention & Research Institute of Texas (CPRIT) Award RR190046 to Y.G. and the NSF CBET-2031242, Welch Foundation (C-1952), and Rice University Startup fund to Y.G. and X.G. We thank the cryo-EM core facility at Baylor College of Medicine and University of Texas McGovern Medical School (both supported by CPRIT RP190602) for sample screening and data collection.

## Author contributions

Y.G. and X.G. conceived the project. X.D., J.Y., E.O., and J.S. performed the cloning. X.D. performed the protein purification, biochemical and structural studies. S.J.O. and X.D. performed the biochemical assays for determining substrate specificity. E.O. performed the cellular assays. All authors analyzed the results, drafted and proofreaded the manuscript.

## Competing interests

The authors declare no competing interests.
