## [Peer Review File · Nature Communications]

Structural basis for the activation of a compact CRISPR-Cas13 nucleaseReviewer #1 (Remarks to the Author):

CRISPR-Cas13 enzymes are RNA-guided RNA endonucleases and have been used for a variety of applications such as RNA knockdown, editing and detection. In this study, Deng et al. biochemically examined RNA substrate preference of a compact Cas13 (Cas13bt3), and determined the cryo-EM structure of Cas13bt3 in complex with a hairpin RNA consisting of a guide RNA and its target RNA. Importantly, a comparison of the present structure (Cas13bt3Act) with the recently reported structures (Cas13bt3Bin and Cas13bt3Int) provided mechanistic insights into Cas13bt3 activation. Furthermore, the authors used the structural information to engineer Cas13bt3 variants with minimal off-target cleavage. Overall, this manuscript is interesting and well written, and it will improve our mechanistic understanding of the diverse CRISPR-Cas13 nucleases and expand the utility of Cas13-based applications. Thus, I would like to support the publication of this manuscript in Nature Communications, after appropriate revision. The authors should consider the following points.

Major points:

L93: "To systemically probe the substrate preference of Cas13bt3, we designed a series of short 10 nt ssRNA reporters each with one nucleotide variation" - Please clarify whether Cas13bt3 cleaves UU (Fig. 1C) or UC (Fig. 1E) in a substrate RNA. To systematically test substrate specificity, it would be better to examine the cleavage activity of Cas13bt3 against U10, A10, G10, and C10, and then those against nnnNNnnnn substrates ("n" is a non-cleaved nucleotide and "NN" is all 16 possible combinations).

Minor points:

L117: "A hairpin substrate by connecting the crRNA and target RNA with a 6 nt (CUUCUU) linker was designed to stabilize the ternary complex" - If you used a crRNA and its target RNA for cryo-EM analysis, please provide this information.

L121: "the dCas13bt3-hpRNA complex showed a good contrast on cryo-EM grids" - Please state the mutations (R84A/R89A/H744A/R739A) introduced into dCas13bt3.

L126: "3D auto refinement yielded a final map at 3.5 Å..." - Did the authors observe a class corresponding to the intermediate state (Cas13bt3Int)?

L165: Please spell out IDL (interdomain linker).

L170: Please change "salt links" to "salt bridges".

L392: "The complex was purified through two steps" - Please explain why the two steps (100 and 400 mM NaCl conditions) were needed for the purification.

Fig. 1E: Please provide the Cas13bt3 concentrations in the legend.

Fig. 5: To clarify the Cas13bt3 activation mechanism, please show the structures of Cas13bt3Bin, Cas13bt3Int, and Cas13bt3Act as ribbon and surface representations with Patches 1 and 2 indicated, to highlight their structural differences. Also, please state in the legend that R84/R89/H744/R739 are the catalytic residues in the HEPN domain.

Fig. S1B: Please provide an explanation of the substrate specificities of RNase T1 and RNase T.

Fig. S1G: Please indicate the analyzed fractions by bars in the chromatograms.

Fig. S3: Please provide the cryo-EM density for the HEPN active site.

Fig. S5: Please explain "bb" (probably, backbone contacts) in the legend.

Please use "prime" rather than "apostrophe" should be used to the RNA ends throughout the

manuscript.

Reviewer #2 (Remarks to the Author):

The type V CRISPR-Cas13 family is RNA-guided RNase, which has been harnessed for RNA targeting and editing. The recently identified Cas13bt3 has robust activity with a smaller size. In this paper, Deng et al solved the cryo-EM structure of Cas13bt3-crRNA-target RNA complex in an activated state, which is distinct from the previously determined structures of Cas13bt3. Therefore, the structure gives authors an opportunity to explore the detailed activation mechanisms of Cas13bt3. Moreover, the authors also designed a variant with comparable on-target activity but reduced trans activity. The paper is well-written, and the data is solid. This paper should be of interest to a broad audience of the CRISPR community, although the structures of Cas13bt3 have been published. Overall, this work represents an important advance in our understanding of the catalytic cycles of the Type V Cas13 family. Nevertheless, there are several issues that should be addressed before publication.

Specific comments:

1. Apart from Cas13a and Cas13d, the authors should compare their structure with Cas13b to give a more comprehensive analysis of Cas13 family, albeit that only the binary structure is available for Cas13b.
2. The overall structural overlay of Act-Cas13bt3 with bin- and Int-Cas13bt3 should be included. What are the conformational changes of other parts when fixing the REC lobe, NUC lobe and duplex, respectively?
3. Another CRISPR effector Cas12i also engages a long spacer/target duplex (~ 30 bp), although Cas12i is an RNA-guided DNase. Interestingly, the longer duplex promotes the catalytic activity of Cas12i, reminiscent of the observation in Cas13bt3. The authors should discuss the similarities and differences between them.
4. The authors performed collateral activity assays to examine the interactions between Cas13bt3 and duplex, and also the binding surface. However, the reduced activity of Cas13bt3 could be caused by either binding or activation or both, the authors should at least discuss these possibilities.
5. Color-blind friendly scheme should be used.

We thank the reviewers for the positive assessment and thoughtful comments and suggestions. We have revised the manuscript accordingly, with itemized responses as below.

Reviewer #1 (Remarks to the Author):

CRISPR-Cas13 enzymes are RNA-guided RNA endonucleases and have been used for a variety of applications such as RNA knockdown, editing and detection. In this study, Deng et al. biochemically examined RNA substrate preference of a compact Cas13 (Cas13bt3), and determined the cryo-EM structure of Cas13bt3 in complex with a hairpin RNA consisting of a guide RNA and its target RNA. Importantly, a comparison of the present structure (Cas13bt3Act) with the recently reported structures (Cas13bt3Bin and Cas13bt3Int) provided mechanistic insights into Cas13bt3 activation. Furthermore, the authors used the structural information to engineer Cas13bt3 variants with minimal off-target cleavage. Overall, this manuscript is interesting and well written, and it will improve our mechanistic understanding of the diverse CRISPR-Cas13 nucleases and expand the utility of Cas13-based applications. Thus, I would like to support the publication of this manuscript in Nature Communications, after appropriate revision. The authors should consider the following points.

Major points:

L93: "To systemically probe the substrate preference of Cas13bt3, we designed a series of short 10 nt ssRNA reporters each with one nucleotide variation" - Please clarify whether Cas13bt3 cleaves UU (Fig. 1C) or UC (Fig. 1E) in a substrate RNA. To systematically test substrate specificity, it would be better to examine the cleavage activity of Cas13bt3 against U10, A10, G10, and C10, and then those against nnnNNnnnn substrates ("n" is a non-cleaved nucleotide and "NN" is all 16 possible combinations).

Response: We thank the reviewer for the suggestion. We have designed and tested A10, G10, C10, and U10 substrates and AAANNAAAAA substrates (where NN refers to any bi-ribonucleotide combination) as suggested. The new results in Fig. S1D reveal that Cas13bt3 cannot cleave A10, G10, and C10 substrates but cleaves U10 with low activity and a pattern of exonuclease cleavage. Furthermore, the experiments in Fig. 1G with AAANNAAAAA substrates unambiguously demonstrate that Cas13bt3 only cleaves at the UC motif, but no other NN motifs.

Minor points:

L117: "A hairpin substrate by connecting the crRNA and target RNA with a 6 nt (CUUCUU) linker was designed to stabilize the ternary complex" - If you used a crRNA and its target RNA for cryo-EM analysis, please provide this information.

Response: In our initial trial of cryo-EM analysis, the Cas13bt3 complex with crRNA and target RNA are used. However, we found that the complex seems disassembled

during cryo-EM sample freezing and no intact protein-RNA complex particles can be found after 2D classification as shown below. Thus, the hairpin substrate was tested, which led to the determination of Cas13bt3 structures with long RNA duplexes.

We provided this information in the manuscript: “To understand the structural basis of Cas13bt3 activation, we attempted to capture Cas13bt3 in a fully activated state with the catalytically inactive Cas13bt3 (dCas13bt3, R84A/H89A/R739A/H744A), crRNA and a 30 nt target. However, the dCas13bt3-crRNA-target ternary complex is unstable during cryo-EM sample freezing and no clear particles with the right size can be found on the cryo-EM grid.”

L121: “the dCas13bt3-hpRNA complex showed a good contrast on cryo-EM grids” - Please state the mutations (R84A/R89A/H744A/R739A) introduced into dCas13bt3.

Response: We have provided detailed information for the mutation at the beginning of the “**Cryo-EM structure of activated Cas13bt3**” section.

L126: “3D auto refinement yielded a final map at 3.5 Å...” - Did the authors observe a class corresponding to the intermediate state (Cas13bt3Int)?

Response: As shown in Fig. S2, we have performed extensive and iterative 3D classifications, but unfortunately, we didn’t observe the intermediate state based on the Ab-initio reconstruction results. We found there may be one class (class 5 in the first round Ab-initio Reconstruction, Fig. S2B) that only contains the REC lobe. But a reasonable high-resolution reconstruction of this class cannot be obtained, possibly due to its small size and flexible nature.

L165: Please spell out IDL (interdomain linker).

Response: We have spelt out the interdomain linker in the abstract and main text.

L170: Please change “salt links” to “salt bridges”.

Response: We have revised “salt links” to “salt bridges”.

L392: “The complex was purified through two steps” - Please explain why the two steps (100 and 400 mM NaCl conditions) were needed for the purification.

Response: In brief, the first gel filtration step was run with high-salt (400 mM NaCl) to exclude any nonspecific protein-RNA complexes, whereas the second gel filtration step

is performed with low-salt (100 mM NaCl) for buffer exchange. Indeed, the dCas13bt3-hpRNA complex at 400 mM NaCl disassembles during cryo-EM freezing (as revealed by 2D classifications below), and we suspect the high salt may reduce the ionic interactions between RNA and protein.

We have added the above information in the method section.

“The complex was purified through two steps of gel filtration chromatography, first at high salt to purify the Cas13bt3 complex with specific hpRNA binding and second at low salt to reduce the salt concentration for stabilizing the ionic interactions between Cas13bt3 and hpRNA (Figure S1H).”

Fig. 1E: Please provide the Cas13bt3 concentrations in the legend.

Response: We have added the Cas13bt3 concentration in the figure legend, which is 50 nM.

Fig. 5: To clarify the Cas13bt3 activation mechanism, please show the structures of Cas13bt3^{Bin}, Cas13bt3^{Int}, and Cas13bt3^{Act} as ribbon and surface representations with Patches 1 and 2 indicated, to highlight their structural differences. Also, please state in the legend that R84/R89/H744/R739 are the catalytic residues in the HEPN domain.

Response: We have prepared new figures showing Patches 1 and 2 in Cas13bt3^{Bin}, Cas13bt3^{Int} and Cas13bt3^{Act} in the new Fig. S7, J-L. As revealed in these figures, Patch1 is only formed in the Cas13bt3^{Act} upon long target RNA binding. We have added in the figure legend that the R84/H89/H744/R739 are catalytic residues as below: “The location of the active site is indicated by the catalytic residues R84A/H89A from HEPN1 and H744A/R739A from HEPN2 in sphere representation.”

Fig. S1B: Please provide an explanation of the substrate specificities of RNase T1 and RNase T.

Response: We have included more explanation of the substrate specificities of RNase T1 and RNase T in the method part. RNase T1 is an endoribonuclease that specifically degrades single-stranded RNA at G residues. RNase T is a single-stranded RNA or DNA-specific nuclease that removes nucleotides in the 3' to 5' direction, but a single 3'-terminal “C” can reduce RNase T action by >100-fold and two consecutive terminal C

residues will stop the enzyme. Moreover, a brief explanation of the control M1 and M2 was also added to Fig. S1 legend:

“The RNaseT1 specifically cleaves ssRNA36 at the G site to produce M1, whereas the RNaseT is a 3'-to-5' exonuclease that will stop at poly C motifs to produce M2. The possible cleavage sites on ssRNA 36 by Cas13bt3 according to the size comparison are indicated and colored in red.”

Fig. S1G: Please indicate the analyzed fractions by bars in the chromatograms.

Response: We added the bars in the chromatograms and explained it in the legend.

Fig. S3: Please provide the cryo-EM density for the HEPN active site.

Response: We provided a new Figure S3D to show the cryo-EM densities around the catalytic HEPN motifs.

Fig. S5: Please explain “bb” (probably, backbone contacts) in the legend.

Response: The “bb” is indicating backbone interaction. It is added in the Fig. S5 legend.

Please use “prime” rather than “apostrophe” should be used to the RNA ends throughout the manuscript.

Response: We have replaced the “apostrophe” with “prime” for all RNA ends.

Reviewer #2 (Remarks to the Author):

The type V CRISPR-Cas13 family is RNA-guided RNase, which has been harnessed for RNA targeting and editing. The recently identified Cas13bt3 has robust activity with a smaller size. In this paper, Deng et al solved the cryo-EM structure of Cas13bt3-crRNA-target RNA complex in an activated state, which is distinct from the previously determined structures of Cas13bt3. Therefore, the structure gives authors an opportunity to explore the detailed activation mechanisms of Cas13bt3. Moreover, the authors also designed a variant with comparable on-target activity but reduced trans activity. The paper is well-written, and the data is solid. This paper should be of interest to a broad audience of the CRISPR community, although the structures of Cas13bt3 have been published. Overall, this work represents an important advance in our understanding of the catalytic cycles of the Type V Cas13 family. Nevertheless, there are several issues that should be addressed before publication.

Specific comments:

1. Apart from Cas13a and Cas13d, the authors should compare their structure with Cas13b to give a more comprehensive analysis of Cas13 family, albeit that only the binary structure is available for Cas13b.

Response: Thanks for the reviewer's suggestion. We have included further analysis of Cas13b structures in Fig. S6, E-H. In brief, the REC lobe of Cas13bt3 aligns well with the REC lobes in the binary structures of PbuCas13b or BzCas13b, whereas the NUC lobes displayed large-scale conformational changes (Fig. S6 E-F). In addition, the HEPN domains in PbuCas13b binary complex are in a closed conformation similar to that in Cas13bt3, whereas HEPN motifs in BzCas13b are further apart (Fig. S6 G-H). We have added the following new sentences in the results:

"In addition, we compared our Cas13bt3 structure to the binary structures of PbuCas13b²³ or BzCas13b²². The REC lobe of Cas13bt3^{Act} aligns well with the REC lobe in both PbuCas13b and BzCas13b, whereas the NUC lobes displayed large-scale conformational changes similarly as observed with respect to Cas13bt3^{Bin} (Figures S6E, S6F). Moreover, the distance between two HEPN motifs in PbuCas13b binary complex²³ is similar to that in Cas13bt3^{Act} (Figure S6G). However, the HEPN motifs are further apart in BzCas13b binary structure²² and additional conformational changes upon target binding are needed to bring them together (Figure S6H)."

2. The overall structural overlay of Act-Cas13bt3 with bin- and Int-Cas13bt3 should be included. What are the conformational changes of other parts when fixing the REC lobe, NUC lobe and duplex, respectively?

Response: To better illustrate the large-scale and complicated conformational changes during Cas13bt3 activation, we provided overall structural comparisons of

Cas13bt3^{Bin}, Cas13bt3^{Int} with Cas13bt3^{Act} by fixing REC lobe, NUC lobe, and RNA duplex in the new figures Fig. S4 G-K, respectively.

3. Another CRISPR effector Cas12i also engages a long spacer/target duplex (~ 30 bp), although Cas12i is an RNA-guided DNase. Interestingly, the longer duplex promotes the catalytic activity of Cas12i, reminiscent of the observation in Cas13bt3. The authors should discuss the similarities and differences between them.

Response: Thank the reviewer for suggesting this similarity. Cas12i is an RNA-guided DNA nickase that exhibit target length-dependent activation. Cas12i undergoes a two-step activation by the propagation of the crRNA-DNA heteroduplex coupled with the dynamic conformational change of a lid motif. However, the conformation change of the lid motif in Cas12i is different from the larger movement of the entire NUC lobe in Cas13bt3. Besides, the seed region of Cas12i is 7 nt segment immediately after the crRNA, whereas the seed region of Cas13bt3 is located at the central region of crRNA-target RNA duplex RNA, suggesting the activation mechanisms of these two effectors are different. We have included citations and new sentences in the discussion.

“Similar target length-dependent activation has been observed in RNA-guided DNA nuclease Cas12i, where Cas12i undergoes a two-step activation by the propagation of the crRNA-DNA heteroduplex coupled with the dynamic conformational change of a lid motif.”

4. The authors performed collateral activity assays to examine the interactions between Cas13bt3 and duplex, and also the binding surface. However, the reduced activity of Cas13bt3 could be caused by either binding or activation or both, the authors should at least discuss these possibilities.

Response: We agree with the reviewer that the reduced collateral activities of the Patch1 mutations are possibly due to the reduced target RNA binding (and the allosteric activation), the reporter RNA binding, or the catalysis. As shown in Figure S8, we found that the target RNA binding affinity is not reduced among the mutations on Patch 1 and Patch 2. Considering the distance of these mutated residues to the catalytic HEPN motifs and the nature of the mutation, we suspect the reduction of activity is more likely due to the reduced binding of reporter RNA.

We have included these new sentences in the discussion:

“Our biochemical data suggested that the positively charged surfaces around the HEPN active site are important for Cas13 activity, which is possibly due to reduced target RNA binding and pairing, reporter/target RNA binding for HEPN cleavage, or catalysis itself. The EMSA binding assays suggested that the Patch 1 and Patch 2 mutations have minimal effect on target RNA binding (Figure S8). Their locations and the charge-reducing/reversal nature of the mutations suggested that these mutations most likely affect Cas13bt3 activities by reducing the reporter/target RNA binding for HEPN nuclease cleavage.”

5. Color-blind friendly scheme should be used.

Response: Thank the reviewer for pointing it out. We have changed the color of IDL2 from red to bright blue in all figures to make it color-blind friendly.

Reviewer #1 (Remarks to the Author):

Deng et al. have addressed my concerns, and the revised manuscript has been improved. Notably, the authors performed in vitro RNA cleavage experiments using various RNA substrates, showing the specific cleavage of the UC motif by Cas13bt3. Therefore, I recommend the publication of this manuscript. Nonetheless, I would like to request the authors to enhance the resolution of figures and improve the clarity of labels in the final version.

Reviewer #2 (Remarks to the Author):

The authors have addressed my concerns, I support the publication of this manuscript.

Reviewer #1 (Remarks to the Author):

Deng et al. have addressed my concerns, and the revised manuscript has been improved. Notably, the authors performed in vitro RNA cleavage experiments using various RNA substrates, showing the specific cleavage of the UC motif by Cas13bt3. Therefore, I recommend the publication of this manuscript. Nonetheless, I would like to request the authors to enhance the resolution of figures and improve the clarity of labels in the final version.

Response: Thanks for the reviewer's thoughtful and thorough review of our manuscript. We are pleased to hear that our revisions have successfully addressed the reviewer's concerns and improved the manuscript. In the revised manuscript, we have enhanced the resolution of the figures and improved the clarity of labels in figures or in the corresponding legend.

Reviewer #2 (Remarks to the Author):

The authors have addressed my concerns, I support the publication of this manuscript.

Response: Thanks for the reviewer reviewing our manuscript and positive feedback. We are pleased to hear that the reviewer's concerns have been addressed and support the publication of our manuscript.